# Learning with Plasticity Rules: Generalization and Robustness

## Abstract

Brains learn robustly, and generalize effortlessly between different learning tasks; in contrast, robustness and generalization across tasks are well known weaknesses of artificial neural nets (ANNs). How can we use our accelerating understanding of the brain to improve these and other aspects of ANNs? Here we hypothesize that (a) Brains employ synaptic plasticity rules that serve as proxies for Gradient Descent (GD); (b) These rules themselves can be learned by GD on the rule parameters; and (c) This process may be a missing ingredient for the development of ANNs that generalize well and are robust to adversarial perturbations. We provide both empirical and theoretical evidence for this hypothesis. In our experiments, plasticity rules for the synaptic weights of recurrent neural nets (RNNs) are learned through GD and are found to perform reasonably well (with no backpropagation). We find that plasticity rules learned by this process generalize from one type of data/classifier to others (e.g., rules learned on synthetic data work well on MNIST/Fashion MNIST) and converge with fewer updates. Moreover, the classifiers learned using plasticity rules exhibit surprising levels of tolerance to adversarial perturbations. In the special case of the last layer of a classification network, we show analytically that GD on the plasticity rule recovers (and improves upon) the perceptron algorithm and the multiplicative weights method. Finally, we argue that applying GD to learning rules is biologically plausible, in the sense that it can be learned over evolutionary time: we describe a genetic setting where natural selection of a numerical parameter over a sequence of generations provably simulates a simple variant of GD.

## 1 Introduction

The brain is the most striking example of a learning device that generalizes robustly across tasks. Artificial neural networks learn specific tasks from labeled examples through backpropagation with formidable accuracy, but generalize quite poorly to a different task, and are brittle under data perturbations. In addition, it is well known that backpropagation is not biorealistic — it cannot be implemented in brains, as it requires the transfer of information from post- to pre-synaptic neurons. This is not, in itself, a disadvantage of backpropagation — *unless one suspects that this lack of biorealism limits ANNs in important dimensions such as cross-task generalization, self-supervision, and robustness.*

We believe that the quest for ANNs that generalize robustly between learning tasks has much inspiration to gain from the study of the way brains work. In this paper we focus on *plasticity rules* (Dayan and Abbott, 2001) — laws controlling changes of the strength of a synapse based on the firing history as seen at the post-synaptic neuron. We provide evidence, both experimental and theoretical, that (a) In the case of RNNs, plasticity rules can successfully replace backpropagation and GD resulting in versatile, generalizable and robust learning; and (b) These rules can be learned efficiently through GD on the rule parameters.

**Plasticity Rules.** *Hebbian learning* ("fire together wire together" Hebb (1949)) is the simplest and most familiar plasticity rule: If there is a synapse $(i, j)$ from neuron $i$ to neuron $j$, and at some point $i$ fires and shortly thereafter $j$ fires, then the synaptic weight of this

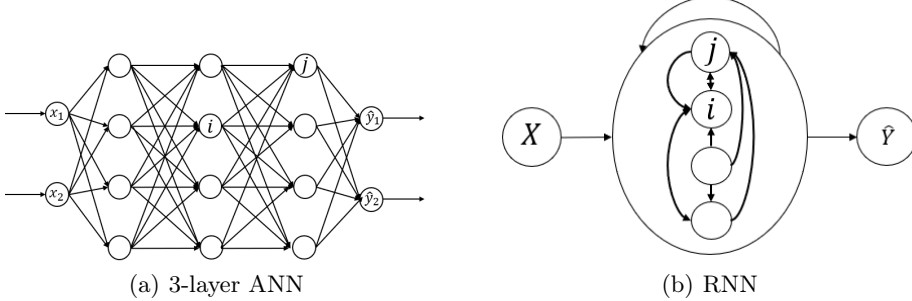

(a) 3-layer ANN                    (b) RNN

Figure 1: Feedforward networks vs RNNs

synapse gets an increment. Over the seven decades since Hebb, many forms of plasticity have been observed experimentally and/or formalized analytically, many of them quite sophisticated and complex, see Dayan and Abbott (2001) for an exposition. All of them dictate a change – increment or decrement – in the synaptic weight of a synapse $(i, j)$ *provided* neurons $i$ and $j$ both fired in some pattern. Intuitively, the decision for the application of a plasticity rule takes place at the post-synaptic neuron $j$, since $j$ receives information from the firing of both $i$ and itself. This is consistent with our understanding of the molecular mechanisms that determine synaptic strength, all of which are complex chemical phenomena taking place at (the dendrite of) $j$.

In this paper we consider plasticity rules as *objects that can be learned.* This fits with the view that existing mechanisms have presumably changed over evolutionary time and are known to differ in their details from one animal species to another. We show experimentally that an RNN can meta-learn a plasticity rule that allows it to learn to perform a classification task *without backpropagation.* This meta-learning is done by GD on the parameters of the rule. Interestingly, the same plasticity rule then performs well on very different tasks and data sets. There are many ways to parameterize a plasticity rule, from full table specifications to small neural networks that take as input observed activation sequence at both ends of a synapse and output the change to the synaptic weight.

**Why RNNs?** RNNs are inspired by, and can model, recurrent activity observed in the brain; they are also especially well-suited to plasticity rules. To illustrate, suppose that we want to train the feed-forward ANN in Figure 1(a) with a plasticity rule. It is clear that the space of possible rules is rather meager. In order to change the weight of link $(i, j)$ after each labeled example, node $j$ will decide the nature of the change based on local information, namely, whether $i$ or $j$ or both fired during this epoch. Thus any learned plasticity rule must be some slight generalization of Hebb's rule[1].

But suppose instead that the three hidden layers have been collapsed into one, resulting in the RNN shown in Figure 1(b), and this collapsed layer fires three times before readout, roughly simulating a feedforward 3-layer network. Now node $j$ knows much more about what happened to link $(i, j)$ during these three epochs; such information was inaccessible in the feedforward setting. Any $2^3 \times 2^3$ matrix of reals is a possible plasticity rule, where $2^3$ is the number of possible firing patterns — such as "fired in the first round, did not fire in the second, fired in the third," or "101" — for each of $i$ and $j$, and the entries of the matrix denote increments/decrements, additive or multiplicative, of the weight of link $(i, j)$. If one updates the entries of this rule by training on a task, it is possible that this rule may be an adequate proxy for the update calculated by backpropagation. Furthermore, we might hope that this rule may even generalize well, performing far above baseline on very different tasks. **Evolution.** We proposed to replace GD in deep learning by biorealistic plasticity rules, and then we use GD to learn the plasticity coefficients. *Are we contradicting ourselves?* After all, the brain did not develop its plasticity rule(s) through GD, but through evolution. But since

---

[1]We could update *all* incoming links to node $j$ based on the firing status of *all of them;* Zenke et al. (2015) suggests that such complex rules may be indeed at work in the animal brain. See the discussion for more on this intriguing research direction.

GD apparently produces good plasticity rules, the question arises, *is evolution at all like GD?* In Section C we address this question analytically. In particular, we prove a general result (which is of some interest by itself in Evolution) stating that the evolution of any real parameter of the phenotype affecting fitness (such as the parameters of the plasticity function) is approximately equivalent to a simple (and suboptimal) variant of GD, as long as the parameter is expressed as the sum of a large number of small genetic contributions (as is known to be the case for many common traits, such as height in humans); the full details are in Appendix C. Hence, it is reasonable to assume that the tuning of such parameters could have been achieved over evolutionary time.

**Summary of Results.** Could such plasticity rules serve as effective learning algorithms? As we show in the following sections, the answer is affirmative: in the special case of the simplest possible network, with no hidden layer and applied to a binary classification task, learning the plasticity rule through GD recovers two classical supervised learning algorithms, the Perceptron algorithm and the Multiplicative Weights (or Winnow) algorithm. We proceed to experiment with learning more complex plasticity rules in a general RNN, establishing that learning plasticity rules leads to performance that is quite good. Even though the performance is not at the same level as ANNs, our experimens show that learning through plasticity has three important benefits: (1) It generalizes well across learning tasks; (2) its convergence to a good classifier is more rapid, i.e., the number of updates (measured by the total number of samples) needed is significantly fewer; and (3), and perhaps more striking, classifiers learned this way appear to be considerably more robust to adversarial perturbations than classifiers learned using GD. An intriguing result here is that the robustness appears to increase significantly with the depth (number of rounds) of the RNN.

## 1.1 Related work

**Plasticity Rules.** Motivated by the brain, learning with plasticity rules has also been studied in machine learning. Early work of Bengio et al. (1990) suggested genetic algorithms for doing so, and later Bengio et al. (1992) explored gradient-based methods as well.Floreano and Urzelai (2000) applied evolving Hebbian plasticity rules to randomly initialized weights for a robot navigation task, while Miconi et al. (2018) introduced *differentiable plasticity* with a plasticity parameter for every edge of a network, which also evolves over time, and applied this to large, high-dimensional data sets. More recently, work by Yaman et al. (2019) is in a similar spirit as ours but with important differences: they apply plasticity rule updates to a specific small 2-layer ANN and find it beneficial; we focus on how rules learned for one task on one network apply to other tasks on other networks, and on the robustness properties of learning through plasticity.

**Other Update Schemes.** There is a variety of mechanisms other than plasticity available to modulate the brain's synaptic weights. Rather than trying to learn more complex plasticity rules, Lillicrap et al. (2020) argue that hand-designed local update rules are sufficient in the presence of feedback connections, and that these are a plausible mechanism for learning in the brain. Whereas backpropagation directly calculates each parameter's contribution to the overall loss, a feedback path with appropriate learned weights can approximate this signal in its stead (such feedback paths are known to be present in the visual cortex). In particular, Metz et al. (2018) learned an update rule which trains both the forward and backward paths and generalizes effectively across tasks, while a body of previous work (see Sacramento et al. (2017); Guerguiev et al. (2017)) has demonstrated that well-known mechanisms from neurobiology can coordinate these forward and backward paths to learn in an online fashion.

Taking a different tack, Wang et al. (2018) train an RNN to implement a general reinforcement learning algorithm, which bears some conceptual similarities to our scheme of learning a general plasticity rule. Here, the meta-learning procedure by which the network's weights are updating is analogous to the action of the dopamine system on the neurons of prefrontal cortex, but when applied to novel tasks the network's weights are frozen. Finally, Andrychowicz et al. (2016) and more recently Maheswaranathan et al. (2020) parameterize a gradient-based optimizer and then optimize these parameters, which is similar in implementation to our strategy for learning plasticity rules.

**Meta-Learning with Evolution.** The plausibility of optimizing meta-learning parameters through evolution has been studied in the literature under the term *neuroevolution*, see Floreano et al. (2008); Stanley et al. (2019). In particular, evolving plasticity rules is a fruitful line of research in its own right (see Soltoggio et al. (2018) for a review), although it uses genetic algorithms to explicitly evolve better architectures and learning rules. Here we use standard models in population genetics to show that the evolution of a numerical parameter can be done in a GD-like fashion.

**Adversarial Robustness.** Lastly, the existence of adversarial perturbations, and in particular learning to avoid them, has been an active topic in recent years, beginning with Goodfellow et al. (2014) and continuing with Madry et al. (2018); Ilyas et al. (2019). Crucially, these methods achieve robust classification by explicitly regularizing the objective function of the network to counter an adversarial attack. We focus on learning methods which by themselves happen to converge to minima that are robust to adversarial perturbations *without explicitly searching for them.*

## 2 Learning (with) Plasticity Rules

Define the RNN plasticity rule $r : \{0,1\}^T \times \{0,1\}^T \to \Re$ to be a function that maps a pair of binary vectors to a real number. The binary vectors correspond to the firing patterns of two neurons $i, j$ connected by a synapse $(i, j)$ in a $T$-round recurrent network. Similarly the output layer plasticity rule is defined by $r_o : \{0,1\}^T \times \{0,1\}$, the binary vector again describing the firing pattern of a neuron, and the $0/1$ value describing whether a node in the output layer corresponds to the true label or not. The functions $r, r_o$ indicate the change to the synapse weight, which can be additive or multiplicative. For example, Hebbian plasticity corresponds to the AND function with $T = 1$. During supervised learning, the plasticity rules are applied independently to each synapse. There are two alternatives here: (1) apply plasticity rules only in the event of *disagreement between the network's output and the true label* of the training example. That is, we assume that, besides the local firing information, the plasticity mechanism also receives a signal about the loss of the current training example; it is known from animal experiments such as Yagishita et al. (2014) that this does happen in the mammalian striatum and cortex through the excretion of dopamine. (2) we apply training rules on all training examples. This requires even lesser coordination, and the time-scales of dopamine action are not an issue. In our experiments, we find that both modes perform equally well (see Fig. 6 in the Appendix). Moreover, the second mode incorporates error information *only at the output layer* (where the correct label is known), making it completely unsupervised throughout the rest of the network. To learn a plasticity rule, we select a model and a dataset to train with, and then randomly initialize a rule. We apply a standard loss function to the output of this network (e.g. cross-entropy loss for classification), but as a function of the parameters of the rule. GD can then be used to update these parameters to minimize the loss function.

**Training.** Our architecture is similar to an RNN. The network consists of an input layer connecting the input to a directed graph $G = (V, E)$, and a fully-connected output layer connecting $G$ to the output nodes. We generate $G$ at random, choosing each edge with probability $p$. Let $A \in \Re^{d \times |V|}$ denote the weights of the input layer, $W \in \Re^{|V| \times |V|}$ the weights of $G$, and $U \in \Re^{|V| \times l}$ the weights of the output layer. Over the course of $T$ rounds, we maintain a hidden vector $h \in \Re^{|V|}$ initialized to zero, and updated as $h \leftarrow c_k(\sigma(W \cdot h + A \cdot x))$ where $x \in \Re^d$ is the input, $\sigma$ is ReLU activation function, and $c_k : \Re^{|V|} \to \Re^{|V|}$ is a notion of a *cap*, a biologically plausible activation function implementing the excitatory-inhibitory balance of a brain area, see Papadimitriou and Vempala (2019). Given a vector $u$, $c_k(u)$ returns a copy of $u$ with only the highest $k$ entries remaining; the rest are set to zero. If at the end of a round $h_i$ is nonzero, we say that the corresponding unit has fired. The output layer consists of linear combinations $U$ of the hidden vector components (one output per label), and a final softmax is then applied. We will refer to this particular architecture as the simple RNN. Given plasticity rules, we train a network as follows. For each individual example in the dataset, we run the forward pass and keep track of the firing sequences of each node. Using these firing sequences, we update the graph using the RNN rule $r$, and the output layer according to $r_o$ as described previously.

**Landscape of rules.** Any function which maps appropriate binary vectors to real numbers defines a rule. An RNN rule can be any function $r : \{0,1\}^T \times \{0,1\}^T \rightarrow \Re$, and the output rule can be any $r_o : \{0,1\}^T \times \{0,1\} \rightarrow \Re$. We consider two different parameterizations: (1) Table: $r$ and $r_o$ are look-up tables of size $2^T \times 2^T$ and $2^T \times 2$, respectively. The entries are the parameters we learn. (2) Small NN: $r$ and $r_o$ are defined by small auxiliary neural networks. These networks take as input the activation sequences, say the concatenation of $s_1, s_2$, and output the update value, $r(s_1, s_2)$. In this case, the weights of the auxiliary network are the parameters we learn.

**Efficiency.** Using tables to represent the plasticity rules is more expressive but requires an exponential number, $(2^T)^2$, of parameters. On the other hand, the complexity of the second method depends only on the size of the auxiliary network, which is independent of the simple RNN size, and its input, the activation sequence, grows linearly as $2T$. We found that training using plasticity rules converges with a significantly smaller number of updates compared to GD. See Fig 2 for a comparison of the two methods.

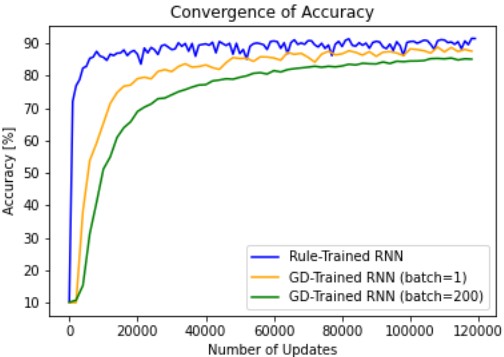

Figure 2: On the standard MNIST data set, we trained the same underlying RNN with $T = 1, |V| = 1000$ with an output layer plasticity rule, and separately with GD (using the standard Adam optimizer, learning rate $10^{-2}$) on the output weights. Note that we did not optimize hyperparameters such as batch size and learning rate. This is only meant to show that plasticity-based training is competitive with gradient methods.

**Data sets.** We use six different datasets. In the first four, 10,000 points are generated from a 10-dimensional normal distribution and assigned binary labels by a linear threshold function (Halfspace), two different ReLU networks each with a single hidden layer of width 1000 and randomly initialized weights (ReLU1 and 2), and a simple RNN with $T = 3, |V| = 100, k = 50$. The last two datasets are the MNIST and Fashion MNIST benchmarks.

## 3 Cross-task Generalization with Plasticity Rules

GD is a general method of optimization, capable of improving the performance of any model for which gradients can be computed. The obvious question is whether plasticity rules offer similarly general strategies for updating the weights of a network. We find that rules learned from simple, low-dimensional datasets generalize to accurately classify data in higher dimensions labeled by much more complex functions, see Fig 3. First, we examine the empirical evidence, and exhibit experiments which demonstrate the remarkable capability of these rules to generalize across tasks. Then we analyze output-layer rules, capturing well-known provable methods for learning linear threshold functions. To test the generalization abilities of these plasticity rules, we learn a rule for a particular network and dataset, and then use it to train other architectures to classify other datasets. In the first experiment, we separately learn output and RNN rules for small networks. With these fixed rules in hand, we then re-train a feedforward and a recurrent network of both small and large sizes on all six of our datasets. The results are clear (see Fig 3); all four models perform well on other datasets, although the large recurrent network consistently outperforms the other

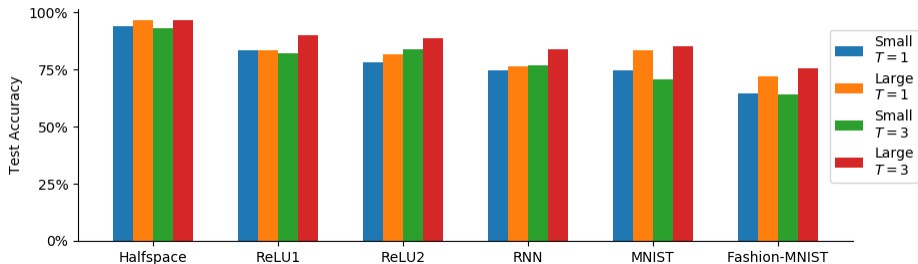

Figure 3: Comparison of various models on different datasets trained using the same plasticity rules. We first learned a plasticity rule for the output weights of a small feedforward network (i.e. $|V| = 100, T = 1$) on the Halfspace dataset, and a plasticity rule for the graph weights of a small recurrent network (i.e. $|V| = 100, T = 3$) on the ReLU1 dataset. We then hold these rules fixed and use them to re-train both of the small models and additionally large models ($|V| = 1000$) on all six datasets, restricting MNIST and Fashion-MNIST to only a random 10,000 training examples in the interest of fair comparison. Averages of 10 re-trainings for each model/dataset combination are shown above.

three models. We have empirically observed a significant increase in accuracy as compared to a network of the same size when using a recurrent network with its weights updated by a plasticity rule (Fig. 3), with the improvement most obvious on the more nonlinear datasets. Significantly, on certain datasets the small recurrent network even outperforms the large feedforward network, suggesting that learned recurrent weights can compensate for fewer neurons. Moreover, a rule learned on one dataset appears to generalize well to others. Thus, it appear that an appropriate RNN plasticity rule represents a general strategy for producing separable representations, although an explanation of how these rules work, let alone whether they are optimal, remains elusive.

## 3.1 ANALYZING THE OUTPUT LAYER PLASTICITY RULE

We first examine update rules for the output layer alone, with the goal of learning a synaptic plasticity rule to update the output layer weights. It is well-known that training just the output layer to minimize well-known loss functions is a convex optimization problem that can be solved efficiently; GD provably works with specialized variants under different assumptions on the data. It has also been established that training just the output layer of a feedforward network, with random weights and a sufficiently wide penultimate layer can provably achieve high classification accuracy (Rahimi and Recht, 2008; Vempala and Wilmes, 2019).

The classical perceptron algorithm for learning an linear threshold function $\ell(x) = \mathrm{sign}(w^* \cdot x)$ is the following iteration, starting with $w = 0$:

$$\text{While there is a misclassified example } x, \quad w \leftarrow w + x\ell(x).$$

This is guaranteed to converge to a halfspace consistent with all the labels in at most $\|w^*\|_2^2 \max_x \|x\|_2^2 / (\min_x \|w^* \cdot x\|)^2$ iterations Rosenblatt (1962); Minsky and Papert (1969). To map this to our setting, we learn a network with a single output neuron, and assume each $x_i \in \{-1, 1\}$. Then this corresponds to the output layer rule in Fig. 1, which depends on the (incorrect) prediction value $p(x) = \mathrm{sign}(w \cdot x)$. This is an additive update rule. The

| Perceptron (additive) | | | MW (multiplicative) | | |
|---|---|---|---|---|---|
| | $p(x) = -1$ | $p(x) = 1$ | | $p(x) = -1$ | $p(x) = 1$ |
| $x_i = -1$ | $-1$ | $1$ | $x_i = 0$ | $1$ | $1$ |
| $x_i = 1$ | $1$ | $-1$ | $x_i = 1$ | $2$ | $\frac{1}{2}$ |

Table 1: The plasticity rules for the Perceptron and MW algorithms

Multiplicative Weights algorithm Littlestone (1987) can be mapped to a similar multiplicative

plasticity rule. Recall that MW only acts on examples where the current hypothesis predicts incorrectly, and then on variables that are "ON", doubling the corresponding weight if the true label is 1, and halving if the true label is $-1$.

Our first theorem is that very similar plasticity rules for the output layer can be automatically discovered in a general setting, i.e., an effective output layer rule can be provably meta-learned.

**Theorem 1.** *GD on an additive output rule, from any starting rule, and network weights initialized to zero, converges to a rule with sign pattern $[-, +; +, -]$.*

In fact, GD provably optimizes the output layer rule. The next theorem follows from the observation that the cross-entropy loss is a convex function of the outer layer weights, which are linear functions of the output layer rule for any fixed graph and sequence of examples.

**Theorem 2.** *The problem of finding the output layer update rule that minimizes the cross entropy loss is a convex optimization problem.*

To explain the generalization itself, we offer a modest (but rigorous) guarantee. In the next section, we will extend this to data that is not perfectly separable.

**Theorem 3.** *Let $r = [-a, a; b, -b]$ be an output layer plasticity rule with $b \geq a > 0$. For data in $\{-1, 1\}^n$ that are strictly linearly separable by a unit vector $w^*$ with $\sum_{i=1}^n w_i^* = 0$, applying this rule to the weights of a linear threshold network converges to a correct classifier.*

## 4    ADVERSARIAL ROBUSTNESS OF LEARNING WITH PLASTICITY RULES

A prevalent attack method is the Fast Gradient Sign Method, first proposed in Goodfellow et al. (2014), which uses the following single step update: $x + \alpha \cdot \text{sign}(\nabla_x L(x, y))$ where $L$ is the loss function, $x$ is the input we wish to perturb, and $y$ is the true label. We use a more powerful adversary, allowing for (1) multiple gradient steps as in Madry et al. (2018), (2) moving directly in the direction of the gradient, instead of using only its sign, as in Rozsa et al. (2016), and (3) targeting a specific class $y'$ that we wish to misclassify the image with:

$$x^{t+1} = \Pi_{x+S}(x^t - \alpha \cdot \nabla_x L(x, y'))$$

where $S$ is the set of allowed perturbations, and $\Pi_{x+S}(v)$ is the projection of a vector $v$ onto the set $x + S$. For a given network and image, we generate nine adversarial images, one for each value of $y' \neq y$. If any of the nine resulting perturbations become misclassified, then we count the original image as misclassified under perturbation (see Appendix A.3 for details). For MNIST, we restrict to perturbations that lie within an $\epsilon$ ball around the original $x$, and to pixel values in the interval $[0, 1]$. We generate an adversarial dataset for both plasticity and gradient trained networks for increasing values of $\epsilon$. Figure 4 shows adversarial images for a rule-trained network need to be considerably more noisy than their GD-trained counterparts.

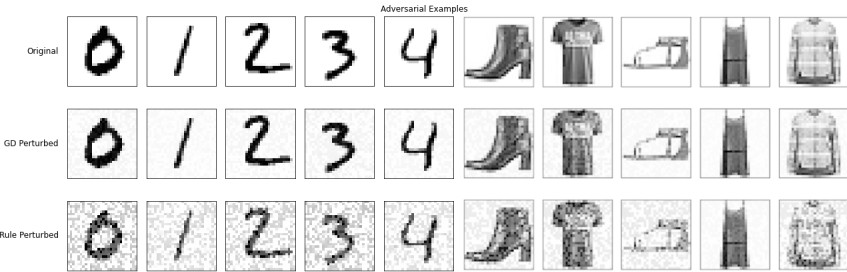

Figure 4: Adversarial perturbations on MNIST (left) and FashionMNIST (right) for a GD-trained network, and a plasticity-trained network. Original images are in the top row.

Figure 5 clearly shows that plasticity rules create more robust classifiers than GD. Madry et al. (2018) explored the relationship between model capacity and adversarial robustness,

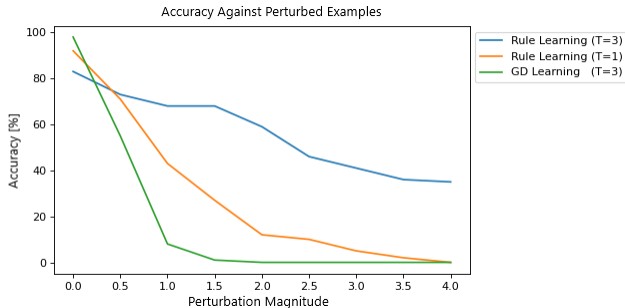

Figure 5: The same network with $|V| = 1000$, cap $= 500$ was trained with plasticity rules for $T = 1, 3$, and with GD. For each trained network, we generate adversarial data-sets with increasing perturbation magnitude.

noting that a larger capacity is needed in order to be robust than to simply classify benign examples. We observe that deeper networks are more robust than shallow ones.

How to explain this robustness? One possible explanation is the following: the RNN finds a rich representation, one in which the examples with different labels can be separated with large margins. More precisely, for most correctly labeled data points, $\epsilon$-balls around them are also classified with the same label. Large margin learning, a celebrated success of Support Vector Machines (Cortes and Vapnik, 1995; Vapnik, 1998), could explain robustness if large margins exist in a suitable kernel space. We show that a similar result holds for a small plasticity based learning, provided we also update on correctly classified examples that are within a small margin of the threshold (see Theorem 4 in the Appendix). The success of RNN plasticity rules in finding a representation amenable to more robust classification is intriguing and merits rigorous explanation.

## 5 DISCUSSION

Learning is the modification of the long-term state of an organism or other system caused by experience; such modification is effected by the system's *learning mechanism.* Meta-learning then must be the structure or parameters of the learning mechanism that remain invariant across learning experiences. In animal brains, synaptic plasticity is just about the only mechanism that qualifies; if meta-learning happens in the animal brain, we propose that it is done through plasticity.

Can these lessons be useful for ANNs? Here we focus on RNNs, because they afford a richer space of synaptic plasticity mechanisms, and we demonstrate that plasticity rules can be learned through GD which (1) achieve reasonably effective learning on a variety of training data without backpropagation; (2) the same rules learned on a data set also perform quite well on new data of a different sort, and on a graph with a different wiring; and (3) these rules can train models which are naturally and significantly more robust to adversarial attacks. Furthermore, in the case of the rules for the output layer, GD produces plasticity rules which recover basic learning algorithms such as the Perceptron and Winnow. We also make the point that learning plasticity rules through GD is biologically plausible, in the sense that learning of any parameter through evolution is, under assumptions, possible through a process which is tantamount to a variant of GD.

We believe that our ideas and results point to a rich and promising field of inquiry. Plasticity in the input layer would probably enhance learning, but would it hurt generalization? Can plasticity rules more complex than the output rule also be dissected analytically? Can plasticity rules work for feed-forward networks? Our observation in Footnote 1 makes this direction worthy of further experimental exploration. Are there ways to *combine* plasticity with backpropagation to enhance generalization while maintaining learning performance? In our experiments, much of the improvement in accuracy is achieved by the output layer, yet learned RNN rules still provide a small but constitent increase in accuracy over random

| # Rounds | MNIST Acc. | Fashion MNIST | Robustness on MNIST |
|---|---|---|---|
| $T = 3$ | 87% | 77% | $\epsilon = 2 : 60\%, \quad \epsilon = 4 : 36\%$ |
| $T = 1^*$ | 93% | 81% | $\epsilon = 2 : 12\%, \quad \epsilon = 4 : 0\%$ |
| $T = 1$ | 85% | 70% | $\epsilon = 2 : 00\%, \quad \epsilon = 4 : 0\%$ |

Table 2: Each experiment uses graphs with $|V| = 1000, k = 500$ and 2 epochs of training. We ran two separate runs for $T = 1$. The starred entry has all entries of the input weights equal to one (normally, we let these be random from a normal distribution). It is unclear why such an initialization produces such a stark improvement in accuracy on MNIST.

fixed weights; we suspect that larger experiments with more than two iterations of the RNN would result in higher accuracy through RNN plasticity. On evolution, are there more general, and more parsimonious, schemes for which evolution is tantamount to GD? Does plasticity also enable *self-supervision* — for example, the creation of powerful representations from unlabeled corpora, as happens both in the language module of the human brain and in modern NLP? Output-layer plasticity can be interpreted in terms of familiar learning algorithms. How can we interpret the much more complex RNN plasticity rules learned? So far, we have been able to only make partial sense out of them. Finally, what is the full range of algorithms that can be realized as synaptic plasticity rules? Does this view, motivated by neural plausibility, yield an interesting complexity-theoretic viewpoint?

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

# A  Experimental Methods

In this section, we give complete details of our experimental procedure. The accompanying code can be found here: `https://github.com/BrainNetwork/BrainNet.git`.

Next, we experimentally show that a rule learned on a specific data set generalizes to new data sets. We learn rules on simple data sets, such as data labelled by a linear threshold function, then use this rule to train a simple RNN on more complex data-sets generated by ReLu networks, simple RNN's, MNIST and Fashion-MNIST.

Then, networks trained by plasticity rules are empirically shown to be more robust than ones trained by GD. Furthermore, as depth increases, so does robustness to adversarial attacks.

Finally, We also describe alternative, arguably more bio-plausible schemes for updating weights during training.

## A.1  Training and Testing Procedure

**Rule-based training.**  First suppose that we are already given output layer plasticity rule $r_o : \{0,1\}^T \times \{0,1\} \to \Re$ and an RNN rule $r : \{0,1\}^T \times \{0,1\}^T \to \Re$. We can now take any simple RNN with $T$ rounds and any data $X = (x^{(1)}, \ldots, x^{(n)})$, and train using these rules. Of course, in the case that $T = 1$, there would be no RNN rule.

1. In the case of additive updates, initialize the graph weights $W$ and output layer weights $U$ to zero. In the case of multiplicative updates, initialize these to 1.

2. For each example $x_i$, perform the forward pass, and keep track of the firing sequence of the nodes.

3. Given the firing sequences of each node, update $W$ according to RNN rule $r$ and $U$, according to output layer rule $r_o$. We scale down the magnitudes of the rule updates by a factor of $\eta$, the step size.

4. The final weights provide the trained simple RNN.

**GD to Learn a Rule.**  We now want to learn a rule specific to a particular data set. For this, we do the following.

1. For each epoch, randomly split the data into batches (we used size 100 or 1000).

2. For each batch, train a network using the current rule as described above.

3. Using the resulting network, compute the cross entropy loss on this batch.

4. Compute the gradient of this loss with resp of choice ect to the parameters of the rules.

5. Update the rules according to the optimizer of choice.

The experiments we have run used the Adam optimizer, with $l_2$ regularization (with a constant of 0.01).

## A.2  Generalization experiments.

We used six different data sets: Halfspace data is labeled by a simple linear threshold function. ReLU1 and ReLU2 data are labeled by two different ReLU feedforward networks, each with a single hidden layer of width 1000 and randomly initialized weights, and two output neurons, and the argmax of the two output neurons was taken to label each example. The simple RNN data was generated by a random simple RNN with $T = 3, |V| = 100, k = 50, p = 0.5$. Each dataset has both training and testing data, each consisting of ten thousand examples. Lastly, we used the standard benchmark MNIST and Fashion-MNIST datasets, with their 28x28 pixel images vectorized to 784 dimensions, where we selected ten thousand random images out of the sixty thousand in each of their training sets.

We began with a simple RNN with $|V| = 100, k = 50, p = 0.5$. Using this network with $T = 1$, we learned an output layer plasticity rule using GD on the Halfspace dataset.

Next, we used a network of the same size with $T = 3$ and ReLU1 data to train, this time learning an RNN plasticity rule parameterized by a single-hidden layer neural network, in addition to the output layer rule.

For each of the two models, we created a new network with a larger graph, $|V| = 1000, k = 500, p = 0.5$. We did not learn new rules specific to these particular graphs, but rather retained the previously learned rules.

Using each model's respective rule(s), we trained the models on the ten thousand training examples from each of the six data sets. Note that this training only consists of initializing the weights of the graph and output layers to 0, and for each misclassified example, update the weights according to the rule, completely without using GD. We did this ten times for each data-set, with the order of examples randomly shuffled each time. We reported the average testing and training accuracy in the figure. In every experiment a learning rate of $\eta = 10^{-2}$ was used, corresponding to weighting the weight update proposed by the rule by a factor of $\eta$.

## A.3    ROBUSTNESS EXPERIMENTS

We generated a simple RNN with $|V| = 1000$ and cap of 500, and trained it separately with plasticity rules and with GD.

We performed two experiments with a rule, one for $T = 1$ and one for $T = 3$. In both cases, the same perceptron-style output rule was used. For $T = 3$, we utilized a small two layer feedforward network to act as the rule. This had a hidden layer of size 20. To train this auxiliary network, we used the method described earlier, however we did so on a smaller simple RNN with $|V| = 200$, and cap of 100.

Once we trained each of the three networks, one hundred random images were chosen to be adversarially perturbed. For a given network and image, we generate nine adversarial images according to the following multi-step attack method previously described, one for each value of $y' \neq y$:

$$x^{t+1} = \Pi_{x+S}(x^t - \alpha \cdot \nabla_x L(x, y'))$$

If any of the nine resulting perturbations become misclassified, then we count the original image as misclassified under perturbation. We perform this process for each $\epsilon = 0.0, 0.5, 1, 1.5, 2.0, 2.5, 3.0, 3.5, 4.0$, allowing a perturbation up to a magnitude of $\epsilon$ in the $l_2$ norm.

## A.4    ALTERNATIVE SCHEMES

**Updates on each example.**    Instead of applying the rules only when we misclassify an example, a more biologically plausible updating scheme would be to perform the rule updates for every example, regardless of the current model's prediction.

Experimentally, this approach has had results very similar to those when updating only for misclassified examples. For instance, Figure 6 is a comparison of the accuracy curves on MNIST when applying the same perceptron update rule on all examples, and on only misclassified examples.

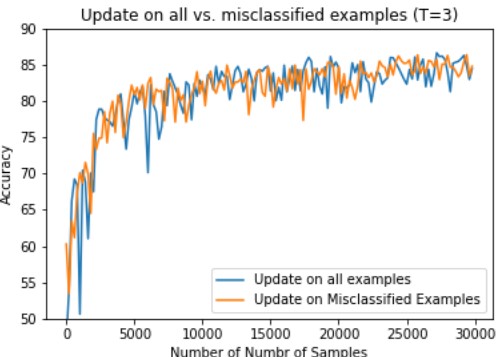

Figure 6: Updating on all examples provides similar results as updating only on misclassified data for a $T = 3, |V| = 1000, k = 500$ simple RNN on MNIST.

**Updates to all edges.**    Our output layer rule only updates edges which lead either to the node corresponding to the true label of the example or to the prediction. Instead, we could apply the rule

to all edges - the first column of the rule indicating the update to the edge leading to the correct label, and the second column indicating the update to the remaining edges. Note that this only affects the multi-label case. For the MNIST data set, we have again had comparable results.

**Using both schemes.** Using both schemes described above (updating at each example, and updating all weights), we provide a computationally efficient analytic solution to finding the optimal output rule (given all else is fixed) with respect to a mean-squared error loss (MSE loss). See Section B.

In the case of our binary classification data, the accuracy we achieve with this optimal rule is comparable to that of our original model, which would update only on misclassified data.

However, when combining both schemes on MNIST data, we begin to see a decline in accuracy. The usual perceptron rule which originally achieved 92% accuracy is now only reaching 88-89%. The optimal rule reached a similar 89% accuracy.

Note that this "optimal" rule is only optimal with respect to the MSE loss, which in general is not particularly well-suited for classification tasks. Additionally, this rule is not necessarily of the same sign pattern

## B  MATHEMATICAL PROOFS

The next theorem is inspired by Freund and Schapire (1999).

**Theorem 4.** *Let $(x_1, y_1), \ldots, (x_m, y_m)$ be a training data set in $\Re^n$ with binary labels such that $\|x\| \leq R$, and $D^2 = \sum_{i=1}^{m} \max\{0, \gamma - y_i(w^* \cdot x)\}$ for some unit vector $w^*$. Suppose we sequentially apply an output layer rule of the form $[-a, a; b, -b]$, with $b \geq a > 0$ to any example whose label is incorrectly predicted or with $\gamma$ of the threshold. Then the number of incorrectly predicted labels is bounded by*

$$O\left(\frac{b^2}{a^2} \cdot \frac{R^2 + D^2 + 2\gamma}{\gamma^2}\right).$$

*Proof of Theorem 1.* For analysis, we assume that we compute the loss after applying the update rule to a random example. For the cross entropy loss, we minimize

$$L(r, W) = \mathbb{E}_{x \sim D}\left(-\log f_{\ell(x)}(r, W, x) \,|\, p(x) \neq \ell(x)\right)$$

Let $\ell_c(x) = 1$ if $\ell(x) = c$ and $\ell_c(x) = 0$ otherwise. $p_c(x)$ is defined similarly for the prediction of $x$. Since the rest of the network is fixed, we can view $L$ and $f$ as functions of just the output layer weight matrix $W$, consisting of weight vectors $w_c$ for each output class $c$. Now $f_c$ is the output neuron value for class $c$, i.e., the result of softmax applied to a linear combination of previous layer outputs. So we have,

$$f_c(r, W, x) = \frac{e^{w_c(r) \cdot y}}{\sum_{c'} e^{w_{c'}(r) \cdot y}}$$

where $y$ is the vector of penultimate layer outputs and $w_c(r)$ is the weight after the rule update, i.e.,

$$w_c(r) \cdot y = \eta \sum_i y_i \sum_{a,b \in \{0,1\}} r(a, b) \Pr(y_i' = a, p_c(x) = b \,|\, p(x) \neq \ell(x)).$$

With $f(z) = e^{z_i} / \sum_j e^{z_j}$, we have

$$\frac{\partial(-\ln f(x))}{\partial z_j} = \frac{\partial}{\partial z_j}(\ln(\sum_k e^{z_k}) - \ln e^{z_i})$$

$$= \frac{e^{z_j}}{\sum_k e^{z_k}} - \chi(i = j).$$

We then compute the gradient of $L$ with respect to $r$:

$$\frac{\partial L}{\partial r(a, b)} = \mathbb{E}_{x \sim D}\left(-\frac{\partial \log f_{\ell(x)}(r, w, x)}{\partial r(a, b)} \,|\, p(x) \neq \ell(x)\right)$$

$$= \mathbb{E}_{x \sim D}\left(\sum_c \left(\frac{\partial w_c(r) \cdot y}{\partial r(a, b)}(f_c(r, W, x) - \chi(c = \ell(x)))\right) \,|\, p(x) \neq \ell(x)\right).$$

In the case of two labels, we get:

$$\frac{\partial L}{\partial r(a,b)} = \mathbb{E}_{x \sim D} \left( f_{\bar{\ell}(x)}(r, W, x) \left( \frac{\partial w_{\bar{\ell}(x)}(r) \cdot y}{\partial r(a,b)} - \frac{\partial w_{\ell(x)}(r) \cdot y}{\partial r(a,b)} \right) \mid p(x) \neq \ell(x) \right).$$

where $\bar{\ell}(x)$ is label opposite $\ell(x)$. Note that

$$\frac{\partial w_c(r) \cdot y}{\partial r(a,b)} = \eta \sum_i \mathbb{E}_{y'}(\chi(y'_i = a, p_c(x) = b, \ell_c(x) \neq b))y_i.$$

Therefore, $\frac{\partial L}{\partial r(a,b)}$ is

$$\eta \mathbb{E}_{x \sim D} \left( f_{\bar{\ell}(x)}(r, W, x) \sum_i y_i \left( \Pr(y'_i = a, p_{\bar{c}}(x) = b \mid \ell_{\bar{c}}(x) \neq b) - \Pr(y'_i = a, p_c(x) = b \mid \ell_c(x) \neq b) \right) \right).$$

From this, we can get the sign of each entry of the rule matrix. First, it is clear that the entries for first and second columns (corresponding to $b = 0, 1$, i.e., the updates to the "correct" and "incorrect" labels) have opposite sign. Next if the gradient for $(0, b)$ is positive, then the gradient for $(1, b)$ is negative, since entries in the second row are negations of the first row minus a positive constant. Then, since we use a standard squared Euclidean norm regularizer, at optimality, the overall gradient is a matrix with the above sign pattern plus the current rule matrix $r$. For this to be zero (at a point with zero gradient), the rule $r$ and the gradient must have the opposite sign pattern. Let $P(a, c) = \Pr(y'_i = a, \ell(x) = c)$. Then, since every $y'_i$ used to update is misclassfied, each coefficient $r(1, 0)$ and multiplier $P(1, \bar{c}) - P(1, c)$ must have the same sign, so if we have $r(1, 0)$ negative, then the $P$ term is negative and the gradient with $a = 0, b = 0$ has positive sign and the rule has negative sign. The signs of the other entries follow similarly. □

*Proof of Theorem 3.* The proof is inspired by the classical proof of the Perceptron algorithm. For data labeled by an unknown linear threshold function $\text{sign}(w^* \cdot x)$ with margin $\gamma$. we consider the invariant $w \cdot w^*/\|w\|_2$. Then on a misclassified example $x$ whose true label is 1, the update is

$$w_i \leftarrow \begin{cases} w_i - a & \text{if } x_i = -1 \\ w_i + b & \text{if } x_i = 1. \end{cases}$$

Therefore the numerator goes from $w^* \cdot w$ to

$$w^* \cdot w - a \sum_{i:x_i=-1} w_i^* + b \sum_{i:x_i=1} w_i^*$$

$$= w^* \cdot w + a(w^* \cdot x) + (b - a) \sum_{i:x_i=1} w_i^*.$$

Then, since $x$ has label 1 we have $-\sum_{i:x_i=-1} w_i^* + \sum_{i:x_i=1} w_i^* \geq \gamma$. Also, by assumption, $\sum_i w_i^* = \sum_{i:x_i=-1} w_i^* + \sum_{i:x_i=1} w_i^* = 0$. Therefore, $\sum_{i:x_i=1} w_i^* \geq \gamma/2$. It follows that the increase in $w^* \cdot w$ in $t$ iterations is at least $ta\gamma$. On the other hand, consider the squared norm of the denominator. After one updated it goes from $\|w\|^2$ to

$$\sum_{i:x_i=-1} (w_i - a)^2 + \sum_{i:x_i=1} (w_i + b)^2$$

$$\leq \|w\|^2 + b^2 n + 2b \sum_{i:x_i=1} w_i - 2a \sum_{i:x_i=-1} w_i$$

$$= \|w\|^2 + b^2 n + 2(b - a) \sum_{i:x_i=1} w_i + 2a(w \cdot x)$$

$$\leq \|w\|^2 + b^2 n + 2(b - a) \sum_{i:x_i=1} w_i$$

where the last step uses the fact that $x$ is misclassified and so $w \cdot x < 0$. We can thus bound the increase in $\|w\|^2$ in $t$ iterations by $Ct$ for some constant $C \leq b^2 n + 2(b - a)\sqrt{n}$. Now since $|w^* \cdot w|/\|w\| \leq 1$, we must have

$$t^2 a^2 \gamma^2 \leq tC$$

or $t < \frac{2b^2 n}{a^2 \gamma^2}$. □

OPTIMAL OUTPUT LAYER RULE FOR MEAN SQUARED ERROR LOSS

We derive an analytic solution to finding the optimal output layer rule given that all else is fixed. By optimal, we mean the rule minimizing the mean-squared error loss of the model after training:

$$L(r) = \frac{1}{n} \sum_{i=1}^{n} \left( \frac{1}{l} \sum_{c=1}^{l} (f_c(r, x^{(i)}) - l_c(x^{(i)}))^2 \right)$$

where we have $n$ data points $x^{(1)} \ldots x^{(n)}$ and $l$ labels. Unlike previously, we do not apply a final softmax to the output, so

$$f_c(r, x^{(i)}) = w_c(r) \cdot y^{(i)}$$

where again $y^{(i)}$ is the vector of penultimate layer outputs corresponding to $x^{(i)}$ and $w_c(r)$ is the final weight vector corresponding to label $c$.

Previously, we have only updated the weights of the output layer if our prediction was incorrect. In this situation, we will instead be updating the weights for every example. Doing so, the final output weights will be independent of the order of the data. Given that the initial weights are initialized at 0, the final weights can be explicitly described by

$$w_c(r) = \eta \sum_{a,b \in \{0,1\}} r(a,b) \sum_{l_c(x^{(i)} \neq b)} \chi^a(y^{(i)})$$

where $\chi^1(y)$ is the standard indicator function. That is $(\chi^1(y))_i = y_i$ if $y_i \neq 0$ and $(\chi^1(y))_i = 0$ otherwise. On the other hand, $(\chi^0(y))_i = 1$ if $y_i = 0$ and 0 otherwise. Note that $y_i$ must be nonnegative since it is the result of a ReLu activation.

For instance, consider the $r(1,0)$, and term contributing to $w_c(r)$:

$$r(1,0) \sum_{l_c(x^{(i)}) \neq c} \chi^1(y^{(i)})$$

Recall that $r(1,0)$ describes the update of an edge $(i,j)$ if node $i$ fired, and node $j$ is the output node corresponding to the true label. We have $l_c(x) \neq 0$ whenever the true label of $x$ is equal to $c$. And, we are updating the weight $(w_c(r))_j$ by $r(1,0)$ whenever the $j^{th}$ node fires, as expected.

Now, we compute the gradient of $L$ with respect to $r$:

$$\frac{\partial L}{\partial r(a,b)} = \frac{2}{n \cdot l} \sum_i \sum_c (w_c(r) \cdot y^{(i)} - l_c(x^{(i)})) \frac{\partial w_c(r) \cdot y^{(i)}}{\partial r(a,b)}$$

Notice that for each $i$ the last term is independent of $r$, and evaluates to a real number:

$$\frac{\partial w_c(r) \cdot y^{(i)}}{\partial r(a,b)} = y^{(i)} \cdot \sum_{l_c(x^{(j)}) \neq b} \chi^a(y^{(j)})$$

Finally, note that the remaining term $w_c(r) \cdot y^{(i)} - l_c(x^{(i)})$ is a linear combination of the entries in $r$ plus some constant. Hence, so must be $\frac{\partial L}{\partial r(a,b)}$.

To find the rule $r$ minimizing the loss, we simply set the gradient to zero. Since each $\frac{\partial L}{\partial r(a,b)} = 0$ is a linear equation in $r$, we have a simple $4 \times 4$ system of linear equations. Its solution is the optimal rule.

Furthermore, it is computationally efficient to determine the optimal rule, taking $O(n \cdot l \cdot d)$ time, where $d$ is the dimension of the penultimate layer, $y$. This can be done by directly computing the $4 \times 4$ linear system as described above. Solving the system afterwards simply takes constant time.

# C  EVOLUTION CAN SIMULATE (A VARIANT OF) GD

We have shown that plasticity rules can be computed though GD in RNNs, and learning is enhanced significantly as a result. On the other hand, plasticity in animals *evolves*. Can we demonstrate analytically that, indeed, plasticity rules can also be learned through evolution? And is there a connection between these two paths on plasticity, namely evolution and GD? Could it be that evolution simulates GD in this case?[2]

Here we show, using the standard mathematical models of population genetics and evolution, that any real parameter such as each of the plasticity coefficients can be adapted by evolution by having such a parameter be the sum of many genetic contributions. This is rather common in genetics — for example, height in mammals seems to be effected additively by over 200 genes, hence the Gaussian nature of height distributions, see Signer-Hasler et al. (2012). Furthermore, we show that the evolution equations ultimately point to GD!

Consider a model in which a haploid organism has $n$ genes $g_1, \ldots, g_n$ each with two alleles $\{+\epsilon, -\epsilon\}$, and suppose that a parameter $Y$ of the phenotype — for example, a coefficient of the plasticity rule — is represented as the sum of these $n$ values. To study the evolution of such organism, consider a sequence of generations indexed by $t$, where at each generation we denote by $x_i^t$ the frequency of allele $i$ in the population, and thus for each individual in the population the expectation of $Y$ is $\bar{Y} = \epsilon \cdot \sum_i (2x_i - 1)$. At each generation, a population is sampled from this distribution, and each individual's performance on the learning task partly determines the individual's fitness — intuitively, its expected number of offspring. We assume that the contribution of this particular parameter to fitness is small — this is reasonable, as there are many other traits contributing to fitness, such as locomotion and digestion. This is known as the *weak selection regime* of evolution Nagylaki (1993); Chastain et al. (2014), and the population genetics equations of how the $x_i$'s (the genetic make-up of the species) evolve are:

$$x_i^{t+1} = \frac{1}{Z^{t+1}}[x_i^t - \theta \cdot (L(\bar{Y}) - L(\bar{Y}_{|+\epsilon}))].$$

(A similar equation holds for the frequency of the other allele, $(1-x_i)^{t-1}$.) $Z^{t+1}$ is a normalizer to be defined soon, $L(Y)$ is the expected loss of the test data when the parameter is $Y$, and $\theta$, assumed to be a very small positive number, is the amount by which aptitude in this learning task will enhance the individual's chance of surviving and procreating. That is, the frequency of the $i$-th gene changes by $\theta$ times the difference between some reference expected loss, taken to be $L(\bar{Y})$, and the expected loss when the $i$-th gene of parameter $Y$ is conditioned to be $+\epsilon$. The function of $Z^{t+1}$ is to keep the allele frequencies adding to one: $Z^{t+1} = 1 + \theta[x_i(L(\bar{Y}) - L(\bar{Y}_{|+\epsilon})) + (1 - x_i)(L(\bar{Y}) - L(\bar{Y}_{|-\epsilon}))]$.

Since $\frac{1}{1+a} = 1 - a + O(a^2)$, the above expression is within $O(\theta^2)$ equal to:

$$x_i^t - \theta \cdot [(L(\bar{Y}) - L(\bar{Y}_{|+\epsilon}))(1 - x_i^2) - (L(\bar{Y}) - L(\bar{Y}_{|-\epsilon}))x_i(1 - x_i)]$$

Now notice that $\bar{Y}_{|\epsilon}$, the expectation of $Y$ conditioned on the value of the gene $i$ being $+\epsilon$, is $(\bar{Y} - \epsilon(1 - 2x_i)) + \epsilon$. To see this, the parenthesis is the expectation of the remaining genes besides gene $i$, and then $\epsilon$ is added to that; and similarly $\bar{Y}_{|-\epsilon} = \bar{Y} - 2\epsilon x$.

Finally, we can approximate the difference $(L(\bar{Y}) - L(\bar{Y} - \epsilon(1 - x_i^2)))$ by $\frac{\partial L}{\partial \bar{Y}}\epsilon(1 - x_i^2) + O(\epsilon^2 \cdot |\frac{\partial^2 L}{\partial \bar{Y}^2}|)$, and similarly for the other difference, to finally obtain, by the chain rule and the fact that $\frac{\partial \bar{Y}}{\partial x_i} = 2\epsilon$,

$$x_i^{t+1} = x_i^t - \theta\frac{\partial L}{\partial x_i^t}(2 - 2x_i^t) + O(\theta^2 + \epsilon^2 \cdot |\frac{\partial^2 L}{\partial \bar{Y}^2}|).$$

Notice now that, ignoring the error term, which is by assumption small, this is GD on gene frequency $x_i$, with the extra factor $2 - 2x_i$, a factor which slows the GD at large values of $x_i$ and accelerates it at small values. Alternatively, this equation is precisely GD on the new variable $z_i = 2x_i - x_i^2$, the integral of the factor $2 - 2x_i$ — note that, appropriately for a variable change, the defining function of $z_i$ is strictly monotone for $x_i$ in $[0, 1]$.

This result holds for the scenario in which each plasticity coefficient is represented by the additive contributions of many genes. What happens in the setting, less wasteful genetically, in which these genes are *shared* between the plasticity coefficients? That is, let us assume that each coefficient is a random linear function of a random subset of these coefficients. That situation is much harder to analyze and compare to GD, but it does work as an effective evolutionary mechanism, see Gorantla et al. (2019), Theorem 1.

---

[2]Recall that Geoff Hinton opined in his Turing award lecture (Hinton, 2019) that "evolution can't get gradients."

