# OpenReview forum: "Learning with Plasticity Rules: Generalization and Robustness"
_ICLR.cc/2021/Conference — Reject_

### Official Review · AnonReviewer4 · 2020-10-25
**Nice new method for meta-training plasticity rules**

**Rating:** 7
**Confidence:** 3

**Review:**

This paper introduces a new method for meta-training plasticity rules, allowing networks to learn new instances of a given domain quickly and efficiently. The method consists in implementing plasticity as an arbitrary function of the past few timesteps of local activity at a synapse (input and output). Experiments show that the method finds reasonably successful rules, that these rules generalize across (some) domains, and that the learning seems more robust to (some) adversarial attacks than plain gradient descent.

While meta-training plasticity rules is not new, I believe the method is novel and quite interesting. I also appreciate the experiments to demonstrate cross-domain generalization and robustness to adversarial attacks.

A possible caveat is that the experiments, though diverse, are still a bit limited. The paper only uses relatively small feedforward networks and recurrent networks with only 3 timesteps. In terms of "real" datasets, only MNIST and Fashion-MNIST are considered. Similarly, the trained plasticity rule seems robust to a certain type of adversarial attack, but is it more robust to other forms of distortion - such as plain noise, deformations, etc. ? I suppose this is tolerable for an introductory paper.

Minor comments:

- Bengio  et al. 1992. “On the Optimization of a Synaptic Learning Rule.”, used gradient descent to meta-learn plasticity rules and should be included in the Related Work section.

- Similarly, Metz et al. 2019: https://openreview.net/forum?id=HkNDsiC9KQ.

- Sometimes, in the figures, it is not clear what exactly is shown. E.g. Which type of plasticity rule is used (network-based or lookup tables)? Also in Figure 4, how exactly are these image generated - what's the criterion for a sufficient error?

Speculative comment: IIUC, the networks in this paper are binary and can be seen as spiking networks.  Interestingly, the space of rules explored by this method seems to include standard models of biological plasticity, such as spike-timing dependent plasticity, as well as more complex "triplet" rules (see e.g. https://www.pnas.org/content/108/48/19383). There's probably some interesting work to be done in this direction (for future work of course, not for this introductory paper).

---

> ### Author Response · Authors · 2020-11-19
> **Response to AnonReviewer4**
>
> Many thanks for the reviewer's comments and insights.
>
> We agree that more extensive experiments would be useful. We are exploring performance on CIFAR-10 (under the time constraints, using unsupervised pre-trained features), and with a necessarily rough first run we are seeing accuracies around 50%.
>
> In terms of robustness, we mainly focused on gradient-based adversarial attacks. However, as you comment, there are many other forms of perturbations to consider. In response, we briefly investigated robustness to perturbations caused by rotation, translation, and adding Gaussian noise to the image. From preliminary results, we saw networks trained with plasiticity rules were more robust to Gaussian noise. For instance, adding noise pixel-wise from a gaussian distribution with mean 0 and variance 0.1, a $T=1$ rule-trained network achieved an accuracy of 65%, while a network with the same architecture trained with gradient descent reached 54%. However, we did not see a significant difference in robustness of plasticity-rule-trained networks (for $T=1,3$) against random rotational or translational perturbations.
>
> Thank you for the references, we have included them (we already had the paper on "Learning a synaptic learning rule" by Bengio-Bengio-Cloutier, but not the later one you point us to). We emphasize that Metz et al. use learned backwards weights to provide an error signal (which would allow for approximate backpropagation), which is a crucial difference from our purely local rules.
>
> We have fixed the figures to be easier to read, thanks for bringing this up. Each image was perturbed adversarially (using GD) till the network gave it a target (incorrect) label or reached the threshold $\epsilon$ distance from the original image. We did this for every target label, and reported the smallest perturbation. The images shown are the resulting minimally distorted images.
>
> Your speculative comment is _very_ interesting --- can STDP be recovered as a learned plasticity rule (it is one, but will GD or some other simple optimization process find it?). We can't wait to run this experiment carefully, as a key next step in this research direction. Thank you!

---

### Official Review · AnonReviewer2 · 2020-10-26
**Well-written, exciting ideas, but lacking in experimental evidence**

**Rating:** 5
**Confidence:** 4

**Review:**

Summary:

This paper uses meta-learning to search for novel, local learning rules in artificial neural networks. This is done by parameterizing local learning rules for feedfoward and recurrent neural networks, then using mini-batch gradient descent to update the parameters of the learning rule.  The authors argue that this is a promising strategy for discovering the learning rules used by biological systems, with three main contributions: (1) they provide proofs that this approach does what we would hope it would do when applied to a single linear layer; (2) experiments demonstrate meta-learning for simple non-linear and recursive architechtures; and (3) the authors provide an argument that evolution could replace gradient descent as the method of searching over possible learning rules. The authors also show through experiments that models trained with these non-gradient methods are more robust to (gradient-based) adversarial attacks.

Overall, I thought this paper was well-written and provided interesting arguments and proofs. However, the experiments are not enough to support the main claims, so I think this paper is borderline.

Pros:

While the main idea of using meta-learning to search for new bio-realistic learning algorithms is not new, the particular formulation used here with recursive neural networks was new as far as I know, and I found this idea very interesting. Most of the work in this area has been focused on feed-forward networks, but as the authors emphasize, recurrent neural networks add a whole new dimension to the space of biologically-plausible, local learning algorithms. An example of such an algorithm is Recirculation, described in Hinton and McClelland 1987, "Learning Representations by Recirculation", which is closely related to Feedback Alignment (Baldi and Sadowski 2018, "Learning in the machine: Recirculation is random backpropagation").

Cons:

My main criticism is that the experiments are not enough to show that the "discovered" learning rule does anything useful in the RNN. As the authors admit, one can achieve good performance in a multi-layer NN by fixing the random weights in the hidden layers and by training only the output layer. I would have really liked to see how the meta-learned algorithm compared to fixed, random representations. The experimental performance of the meta-learned algorithm on MNIST is quite poor (~80% test accuracy, Figure 3), so its unclear what is going on.

The results of the adversarial robustness experiments are not surprising. Adding additive or multiplicative noise during training will also make the trained networks more robust. It think these experiments actually distract from the main ideas of the paper. It would have been better to more carefully explore whether the learning algorithm can learn more difficult functions.

Section 2 describes two possible alternatives for the plasticity rules: one that incorporates information about the error, and one that does not. I think it is important to highlight the fact that the latter is an unsupervised plasticity rule --- the meta-learning algorithm has access to the target output, but the local plasticity rule does not. So while a plasticity rule trained on Dataset 1 does have some information about the Dataset 1 targets (by way of the meta-learning updates), when it is applied to Dataset 2 it never receives any information about Dataset 2's target, and is thus unsupervised. This is an important distinction between the two approaches.

---

> ### Author Response · Authors · 2020-11-19
> **Response to AnonReviewer2**
>
> Many thanks for the reviewer's careful and encouraging review.
>
> In our experiments, using an RNN rule consistently offers an improvement of a few percent in accuracy. We indeed had the results of the experiment the reviewer suggested (fixed random weights for the RNN, train only output layer), and using an RNN layer is always a little better. We didn't report this earlier, but have now included it in the discussion. As we say, we do not know how to rigorously explain this improvement, and consider this a major open problem. We also note that using the RNN rule is critical to the improvement in adversarial robustness (the difference is much greater).
>
> On the robustness findings, we emphasize that we do not do anything special to counteract the adversarial perturbations. Our networks are trained, with plasticity rules, on the original unperturbed data sets, and still they turn out to be quite robust to adversarial perturbations. This is unlike much of the work in this field, where the training itself is modified.
>
> Thank you for the suggestion on highlighting the unsupervised aspect of the plasticity rules. We added this point to Section 2.

---

### Official Review · AnonReviewer3 · 2020-10-29
**Learning plasticity rules**

**Rating:** 7
**Confidence:** 3

**Review:**

Summary:

This is a fascinating paper which push the bounds of artificial neural network knowledge and understanding, while questioning the typical approach of considering various neural networks branches in isolation. Concretely, it shows that by using concepts from neuroevolution together with deep learning concepts, we can learn how to learn learning rules (i.e. plasticity rules) which can generalize across tasks and can train neural networks which are more robust to adversarial attacks than typical networks trained with stochastic gradient descent.

Strong points:

•	In my opinion, the paper is visionary. It answers few questions, while opening the path for a large number of new research directions and new unanswered questions.

•	The paper has a very well-balanced content of novelty, math, computer science, neuroscience, and even philosophy.

•	The paper is very well written and anchored in a multidisciplinary literature. It has the potential of becoming a “must read” paper in the future.


Weak points:

•	The datasets used, including MNIST and Fashion-MNIST are rather simple. It would be very interesting to see how the approach behaves on more complex datasets.


During the discussion phase, I would recommend to the authors to address the following comments:

1. In the limit of time, try to perform experiments also on CIFAR 10/100. I believe that it would be interesting to see on CIFAR 100, the behavior of three types of learned plasticity rules: (1) plasticity rules learned on the simple datasets, (2) plasticity rules learned on CIFAR 10, and (3) plasticity rules learned on a subset of the CIFAR 100 training set.

2. Are you encountering problems with ReLU activation in recurrent networks such as exploding or vanishing weights? Does your approach work also with hyperbolic tangent?

3. I believe that it would help the paper clarity if you can add a table towards the end of the paper to summarise the main results in terms of accuracy, training time, etc.

4. Perform a proof-read of the whole paper to improve the English usage and the presentation. For instance: typos (e.g. “rule, The next theorem”), unit measures for axis labels (e.g. figure 5 – accuracy [%]), etc.

---

> ### Author Response · Authors · 2020-11-19
> **Response to AnonReviewer3**
>
> Thank you for the encouraging review! We do plan to continue this line of work, and your positive comments and suggestions help a lot.
> 1. It is certainly natural to test plasticity-based learning on more complex data sets such as CIFAR 10/100. One complication is that for such networks, good baseline performance seems to require convolutional layers and considerable depth. To focus on the benefits of plasticity rules, we decided to learn on top of an unsupervised set of features, namely the publicly available ResNet50 trained via MOCO. The experiment requires a bit more fine-tuning but we have seen accuracies around 50%.
> 2. We have not had any issues with weights exploding or vanishing. We believe this is partly because of our use of a $k$-cap (top $k$ fire), rather than individual thresholds. We should also note that the number of rounds of our RNN is relatively small, so such an effect might not be significant in any case. We are currently extending to greater depth and will keep track of this.
> 3. Thank you, we included such a summary in the discussion.
> 4. We have tried to correct all typos.

---

### Official Review · AnonReviewer1 · 2020-11-02
**Learning to learn... a reboot of an old idea.**

**Rating:** 4
**Confidence:** 4

**Review:**

The authors propose to encapsulate the update rule for a neural net into a look-up table specifying weight changes for each combination of "pre-synaptic" input to the weight, and "post-synaptic" activation of the unit receiving that incident connection. They learn the elements of this matrix by gradient descent, and then use that learned update rule to train neural nets on a new task. This is motivated by a separation of timescales biologically, wherein learning rules might be evolved over long timescales, and then act within each brain over shorter ones.

There is a nice discussion of related previous work, but it misses a few key items that, to me, diminish somewhat the novelty of this work. That's okay: being first isn't everything. But I think it is important to point out to readers what is new and better about this work vs previous work.

a) The auto ML zero paper from Quoc Le et al. (arXiv 2003.03384). They learn both architectures and learning rules via simulated evolution

b) Andrychowicz, M., Denil, M., Gomez, S., Hoffman, M. W., Pfau, D., Schaul, T., ... & De Freitas, N. (2016). Learning to learn by gradient descent by gradient descent. In Advances in neural information processing systems (pp. 3981-3989). They use GD to learn plasticity rules.

c) A few recent papers on bio-plausible backprop-type algorithms.
i)  Burst-dependent synaptic plasticity can coordinate learning in hierarchical circuits. Biorxiv: https://doi.org/10.1101/2020.03.30.015511 from Naud and colleagues
ii) Guerguiev, Jordan, Timothy P. Lillicrap, and Blake A. Richards. "Towards deep learning with segregated dendrites." ELife 6 (2017): e22901.
iii) Sacramento, João, Rui Ponte Costa, Yoshua Bengio, and Walter Senn. "Dendritic cortical microcircuits approximate the backpropagation algorithm." In Advances in neural information processing systems, pp. 8721-8732. 2018.

Aside from the relation to prior work, I have a few technical and conceptual questions / comments:

1) Fig. 2: were all three nets given the same initialization? That could matter for comparing the training curves of accuracy vs. training time because a good initialization could give one learning rule an apparent advantage. And given the accuracy at t=0, it doesn't look like they are the same.

2) I like that the authors studied generalization of the learned rule between tasks: that is important (although, SGD also generalizes well). I'm a bit less impressed by the performance obtained in the MNIST and fashion MNIST tasks. At the same time, using two-factor rules (update is a function just of pre- and post-synaptic inputs) to solve MNIST sounds hard given that there's no credit assignment signal. I think that the authors would be well served to read up on the papers on bioplausible deep learning, and consider variants of this work that include a credit assignment signal.

---

> ### Author Response · Authors · 2020-11-19
> **Response to AnonReviewer1**
>
> We thank the reviewer for putting our work in the context of the broader field of learning-to-learn, and for the specific references. The first reference (and earlier work) is about automating the discovery of ML algorithms in a structured programming framework. The second is about optimizing design choices for gradient-based optimization algorithms (e.g., parameters of gradient descent). We are happy to include these references. However, we want to emphasize that our method does not rely on evolution explicitly: we optimize our plasticity rules using GD; we also argue that these types of changes could happen in real neural systems via evolution.
> We are not by any means pioneers of this general approach (we discuss several related papers). Rather our contribution is in the study of learning with plasticity rules and its benefits; we find that learning plasticity rules is surprisingly versatile and robust. This, we believe, is both new and interesting, in part because of the widely accepted bio-plausibility of learning with synaptic rules. On the other hand, there is an ongoing debate about the bio-plausibility of GD, and specifically whether it happens during learning in the brain. We have referenced multiple papers in this line, including a very recent survey by Lillicrap et al ("Backprop in the brain", Nature Reviews Neuroscience, 2020) which builds upon many schemes blending backprop and plasticity, including the ones you recommended. We have added the papers mentioned in your review and we welcome other suggestions.
>
> To address technical questions/comments:
> 1. Yes, all 3 nets were initialized identically. We did not display the 0'th epoch earlier, but we do that now.
> 2. We agree that SGD is a general algorithm, but we thought it is surprising that plasticity rules, which appear much more limited, also generalize across tasks and data sets, and are more robust to adversarial perturbations. Thank you for the excellent suggestion to consider bioplausible ways of credit assignment.

---

### Author Response · Authors · 2020-11-23
**Experiments with CIFAR-10**

A natural question is how well plasticity-based learning works for more complex datasets such as CIFAR-10. For these data sets, using a deep convolutional network seems to be important for decent accuracy. To focus on plasticity-based learning, we trained classifiers using the RNN-based architectures that we employed for our earlier experiments on top of features generated from pretrained CNNs. One network was the publicly-available weights for ResNet50 which were learned without labels via MoCov2 using ImageNet; this provides 2048 features, of which we selected the 512 highest variance features. The other network was SqueezeNet v1.1 which we trained with supervision on CIFAR-10, and it provides 512 features by default. Our classifiers were a single layer, two layer, and RNN+output layer network, where the multi-layer networks had 1024 hidden units. For plasticity-based training, we used the same output and graph rules previously learned on MNIST.

We found that the supervised features offered better performance, of between 69-71% accuracy for all combinations of architecture and training scheme. (The supervised network achieved a test accuracy of 71% during pre-training.) The unsupervised features were harder to learn from: the single-layer GD-trained network achieved 55% accuracy, the single-layer plasticity-trained network achieved 44%, and all others were between 33-35%. For both sets of features, a small amount of perturbation had more of an impact than we observed on MNIST, but, notably, the recurrent network trained with plasticity maintained consistently higher accuracy against larger adversarial perturbations, while the performance of other architectures decreased.

In summary, in preliminary experiments using the same architecture on top of unsupervised (MoCo) or supervised (SqueezeNet) features, led to decent accuracy, and clearly indicated that RNNs trained with plasticity are more robust to adversarial perturbations, agreeing with our earlier findings.


__Sources:__

He et al, "Momentum Contrast for Unsupervised Visual Representation Learning", 2019.

Iandola et al, "SqueezeNet: AlexNet-level accuracy with 50x fewer parameters and <0.5MB model size", 2016.

---

### Decision · Program_Chairs · 2021-01-07
**Final Decision**

**Decision:**

Reject

**Comment:**

This paper explores meta-learning of local plasticity rules for ANNs. The authors demonstrate that they can meta-learn purely local learning rules that can generalize from one dataset to another (though with fairly low performance, it should be noted), and they provide some data suggesting that these rules lead to more robustness to adversarial images. The reviews were mixed, but some of the reviewers were very positive about it. Specifically, there are the following nice aspects of this work:

A) The meta-learning scheme has interesting potential for capturing/learning biological plasticity rules, since it operates on binary sequences, which appears to be a novel approach that could help to explain things like STDP rules.

B) It is encouraging to see that the learning rules can generalise to new tasks, even if the performance isn't great.

C) The authors provide some interesting analytical results on convergence of the rules for the output layer.

However, the paper suffers from some significant issues:

1) The authors do not adequately evaluate the learned rules. Specifically:

- The comparison to GD in Fig. 2 is not providing an accurate reflection of GD learning capabilities, since a simple delta rule applied directly to pixels can achieve better than 90% accuracy on MNIST. Thus, the claim that the learned rules are "competitive with GD" is clearly false.

- The authors do not compare to any unsupervised learning rules, despite the fact that the recurrent rules are not receiving information about the labels, and are thus really a form of unsupervised learning.

- There are almost no results regarding the nature of the recurrent rules that are learned, either experimental or analytical. Given positive point (A) above, this is particularly unfortunate and misses a potential key insight for the paper.

2) The authors do not situate their work adequately within the meta-learning for biologically plausible rules field. There are no experimental comparisons to any other meta-learning approaches herein. Moreover, they do not compare to any known biological rules, nor papers that attempt to meta-learn them. Specifically, several papers have come out in recent years that should be compared to here:

https://proceedings.neurips.cc/paper/2020/file/f291e10ec3263bd7724556d62e70e25d-Paper.pdf https://www.biorxiv.org/content/10.1101/2019.12.30.891184v1.full.pdf https://proceedings.neurips.cc/paper/2020/file/bdbd5ebfde4934142c8a88e7a3796cd5-Paper.pdf https://openreview.net/pdf?id=HJlKNmFIUB https://proceedings.neurips.cc/paper/2020/file/ee23e7ad9b473ad072d57aaa9b2a5222-Paper.pdf

And, the authors should consider examining the rules that are learned and how they compare to biological rules (e.g. forms of STDP), if indeed biological insights are the primary goal.

3) The paper needs to provide better motivation and analyses for the robustness results. Why explore robustness? What is the hypothesis about why these meta-learned rules may provide better robustness? There is little motivation provided. Also, the authors provide very little insight into why you achieved better robustness and insufficient experimental details for readers to even infer this. This section requires far more work to provide any kind of meaningful insight to a reader. What was the nature of the representations learned? How are they different from GD learned representations? Was it related to the ideas in Theorem 4? Note: Theorem 4 is interesting, but only applies to a specific form of output rule.

4) In general, the motivations and clarity of the paper need a lot of work. What are the authors hoping to achieve? Biological insights? Then do some analyses and comparisons to biology. More robust and generalisable ML? Then do more rigorous evaluations of performance and comparisons to other ML techniques. Some combination of both? Then make the mixed target much clearer.

5) The authors need to tidy up the paper substantially, and do better at connecting the theorems to the rest of the paper, particularly for the last 2 theorems in the appendix. Also, note, Theorems 2 & 4 appear to have no proofs.

Given the above considerations, the AC does not feel that this paper is ready for publication. This decision was reached after some discussion with the reviewers. But, the AC and the reviewers want to encourage the authors to take these comments on board to improve their paper for future submissions, as the paper is not without merit.